# Understanding DDPM Latent Codes Through Optimal Transport

**Valentin Khrulkov**
Yandex
Moscow, Russia
khrulkov.v@gmail.com

**Gleb Ryzhakov & Andrei Chertkov**
Skolkovo Institute of Science and Technology
Moscow, Russia
{a.chertkov,g.ryzhakov}@skoltech.ru

**Ivan Oseledets**
Skolkovo Institute of Science and Technology and AIRI
Moscow, Russia
i.oseledets@skoltech.ru

## Abstract

Diffusion models have recently outperformed alternative approaches to model the distribution of natural images. Such diffusion models allow for deterministic sampling via the probability flow ODE, giving rise to a latent space and an encoder map. While having important practical applications, such as the estimation of the likelihood, the theoretical properties of this map are not yet fully understood. In the present work, we partially address this question for the popular case of the VP-SDE (DDPM) approach. We show that, perhaps surprisingly, the DDPM encoder map coincides with the optimal transport map for common distributions; we support this hypothesis by extensive numerical experiments using advanced tensor train solver for multidimensional Fokker-Planck equation. We provide additional theoretical evidence for the case of multivariate normal distributions.

## 1 Introduction

Denoising diffusion probabilistic models (DDPMs) (Sohl-Dickstein et al., 2015; Ho et al., 2020) have recently outperformed alternative approaches to model the distribution of natural images both in the realism of individual samples and their diversity (Dhariwal & Nichol, 2021). These advantages of diffusion models are successfully exploited in applications, such as colorization (Song et al., 2021b), inpainting (Song et al., 2021b), super-resolution (Saharia et al., 2021; Li et al., 2022), and semantic editing (Meng et al., 2021), where DDPM often achieve more impressive results compared to GANs.

One crucial feature of diffusion models is the existence of a deterministic invertible mapping from the data distribution to the limiting distribution of the diffusion process, commonly being a standard normal distribution. This approach termed denoising diffusion implicit model (DDIM) (Song et al., 2021a), or the probability flow ODE in the continuous model (Song et al., 2021b), allows to invert real images easily, perform data manipulations, obtain a uniquely identifiable encoding, as well as to compute exact likelihoods. Despite these appealing features, not much is known about the actual mathematical properties of the encoder map and corresponding latent codes, which is the question we address in this work. Concretely, in this paper, we show that for the DDPM diffusion process, based on the Variance Preserving (VP) SDE (Song et al., 2021b), this encoder map with a large numerical accuracy coincides with the *Monge optimal transport map* between the data distribution and the standard normal distribution. We provide extensive empirical evidence on controlled synthetic examples and real datasets and give a proof of equality for the case of multivariate normal distributions. Our findings suggest a complete description of the encoder map and an intuitive approach to understanding the 'structure' of a latent code for DDPMs trained on visual data. In this case, the pixel–based Euclidean distance corresponds to high–level texture and color–level similarity, directly observed on real DDPMs. To summarize, the contributions of our paper are:

1. We theoretically verify that for the case of multivariate normal distributions the Monge optimal transport map coincides with the DDPM encoder map.

2. We study the DDPM encoder map by numerically solving the Fokker-Planck equation on a large class of synthetic distributions and show that the equality holds up to negligible errors.

3. We provide additional qualitative empirical evidence supporting our hypothesis on real image datasets.

## 2 REMINDER ON DIFFUSION MODELS

We start by recalling various concepts from the theory of diffusion models and stochastic differential equations (SDEs).

### 2.1 DENOISING DIFFUSION PROBABILISTIC MODELS

Denoising diffusion probabilistic models (DDPMs) is a class of generative models recently shown to obtain excellent performance on the task of image synthesis (Dhariwal & Nichol, 2021; Ho et al., 2020; Song et al., 2021b). We start with a forward (non-parametric) diffusion which gradually adds noise to data, transforming it into a Gaussian distribution. Formally, we specify the transitions probabilities as

$$q(x_t|x_{t-1}) \coloneqq \mathcal{N}(x_t; \sqrt{1-\beta_t}x_{t-1}, \beta_t I), \tag{1}$$

for some fixed variance schedule $\beta_1, \ldots, \beta_t$. Importantly, a noisy sample $x_t$ can be obtained directly from the clean sample $x_0$ as $x_t = \sqrt{\bar{\alpha}_t}x_0 + \sqrt{1-\bar{\alpha}_t}\mathbf{z}$, with $\mathbf{z} \sim \mathcal{N}(0, I)$ and $\alpha_t \coloneqq 1-\beta_t$, $\bar{\alpha}_t \coloneqq \prod_{s=1}^{t} \alpha_s$. The generative model then learns to reverse this process and thus gradually produce realistic samples from noise. Specifically, DDPM learns parameterized Gaussian transitions:

$$p_\theta(x_{t-1}|x_t) \coloneqq \mathcal{N}(x_t; a_\theta(x_t, t), \sigma_t). \tag{2}$$

In practice, rather than predicting the mean of the distribution in Equation (2), the noise predictor network $\epsilon_\theta(x_t, t)$ predicts the noise component from the sample $x_t$ and the step $t$; the mean is then a linear combination of this noise component and $x_t$. The covariances $\sigma_t$ can be either fixed or learned as well; the latter was shown to improve the quality of models (Nichol & Dhariwal, 2021). Interestingly, the noise predictor $\epsilon_\theta(x_t, t)$ is tightly related to the *score function* (Stein, 1972; Liu et al., 2016; Gorham, 2017) of the intermediate distributions; specifically by defining $s_\theta(x_t, t)$ as

$$s_\theta(x_t, t) = \frac{\epsilon_\theta(x_t, t)}{\sqrt{1-\bar{\alpha}_t}} \tag{3}$$

we obtain that $s_{\theta^*}(x_t, t) \approx \nabla_x \log p_t(x_t)$, with $p_t(x_t)$ being the density of the target distribution after $t$ steps of the diffusion process and $\theta^*$ are the parameters at convergence.

**Stochastic and deterministic sampling.** Given the DDPM model, the generative process is expressed as

$$x_{t-1} = \frac{1}{\sqrt{\alpha_t}}\left(x_t - \frac{1-\alpha_t}{\sqrt{1-\bar{\alpha}_t}}\epsilon_\theta(x_t, t)\right) + \sigma_t\mathbf{z},$$

with the initial sample $x_T \sim \mathcal{N}(0, I)$ and $\mathbf{z} \sim \mathcal{N}(0, I)$. The sampling procedure is stochastic, and no single 'latent space' exists. The authors of Song et al. (2021a) proposed a deterministic approach to produce samples from the target distribution, termed DDIM (denoising diffusion implicit model). Importantly, this approach does not require retraining DDPM and only changes the sampling algorithm; the obtained marginal probability distributions $p_t$ are equal to those produced by the stochastic sampling. It takes the following form:

$$x_{t-1} = \sqrt{\bar{\alpha}_{t-1}}\left(\frac{x_t - \sqrt{1-\bar{\alpha}_t}\epsilon_\theta(x_t, t)}{\sqrt{\bar{\alpha}_t}}\right) + \sqrt{1-\bar{\alpha}_{t-1}} \cdot \epsilon_\theta(x_t, t). \tag{4}$$

By utilizing DDIM, we obtain a concept of a latent space and encoder for diffusion models, since the only input to the generative model now is $x_T \sim \mathcal{N}(0, I)$ for sufficiently large $T$. In the next section, we will see how it is defined in the continuous setup.

## 2.2 SDE VIEW ON DIFFUSION MODELS

The authors of Meng et al. (2021) proposed to view diffusion models as a discretization of certain stochastic differential equations. SDEs generalize standard ordinary differential equations (ODEs) by injecting random noise into dynamics. Specifically, the diffusion process specified by Equation (1) is a discretization of the following SDE:

$$dx = -\frac{1}{2}\beta(t)x dt + \sqrt{\beta(t)} dw. \tag{5}$$

where $dx$ is an increment of $x$ over the infinitesimal time step $dt$. Here, $w$ represents a Brownian motion process, so $dw$ can be intuitively understood as an infinitesimal Gaussian noise. Similar to DDPMs, an initial distribution with density $p_0$ evolves into a standard normal distribution under this SDE. Various DDPM algorithms can be seen as discretizations of SDE-based algorithms. For instance, the continuous analogue of DDIM sampling (4) is constructed in the following way. The following (deterministic) ODE, termed *probability flow ODE* (Song et al., 2021b)

$$dx = -\frac{\beta(t)}{2}\left[x + \nabla_x \log p_t(x)\right] dt, \tag{6}$$

results in the same marginal distributions $p_t$ as given by the SDE (5). Here, the score function $\nabla_x \log p_t(x)$ is similarly approximated via a neural network during training. If we reverse this ODE, we obtain a mapping from the limiting distribution to the data distribution. It turns out that the DDIM sampling is a discretization of this continuous approach (Song et al., 2021b).

The well-developed theory of SDEs provides us with a rigorous mathematical apparatus allowing us to study the properties of diffusion models at hand. We will utilize the following well-known equation termed the forward Kolmogorov, or Fokker-Planck equation (Risken, 1996). This equation determines how a density $p_0$ evolves under a (general) SDE. For (5) it takes the following form:

$$\frac{\partial p_t}{\partial t} = \frac{\beta(t)}{2}\left[\nabla_x \cdot (x p_t) + \nabla_x^2 p_t\right]. \tag{7}$$

Note, that $\beta(t)$ can be removed from (7) by change of variables $dt := \frac{\beta(t)}{2} dt$, resulting in the PDE of the form

$$\frac{\partial p_t}{\partial t} = \nabla_x \cdot (x p_t) + \nabla_x^2 p_t. \tag{8}$$

Without loss of generality, we will work with this simplified equation since the specific form of $\beta(t)$ does not affect the results. In this case, the probability flow ODE takes the following form:

$$dx = -\left[x + \nabla_x \log p_t(x)\right] dt. \tag{9}$$

**Encoder map.**    As mentioned above, by considering the probability flow ODE (6), we can obtain latent codes for samples from the original distribution. For a given distribution $\mu_0$ and a timestep $t$, let us denote the flow generated by this vector field as $E_{\mu_0}(t, \cdot)$. I.e., a point $x \sim \mu_0$ is mapped to $E_{\mu_0}(t, x)$ when transported along the vector field for a time $t$. The 'final' encoding map is obtained when $t \to \infty$, i.e,

$$E_{\mu_0}(x) := \lim_{t \to \infty} E_{\mu_0}(t, x).$$

Note that $E_{\mu_0}$ implicitly depends on all the intermediate densities $\mu_t$ obtained from the diffusion process (or the Fokker-Planck equation).

## 3 DDPM ENCODER AND OPTIMAL TRANSPORT

By construction and properties of the phase flow, the map $E_\mu$ transforms the original distribution $\mu_0$ into the standard normal distribution $\pi := \mu_\infty \sim \mathcal{N}(0, I)$. Our further analysis investigates a perhaps surprising hypothesis that this map is very close to the *optimal transport* map. How to arrive at this idea? A priori, it seems nontrivial that the theory of optimal transportation is related to the Fokker-Planck equation and diffusion processes. Some intuition may be obtained from the Otto calculus (Otto, 1996; 2001; Villani, 2009). It presents an alternative view of the solutions of the Fokker-Planck-type PDEs and diffusion equations in the following manner. It can be shown that the trajectory $\{\mu_t\}_{t=0}^\infty$, obtained by solving the Fokker-Planck equation (or an equivalent diffusion SDE),

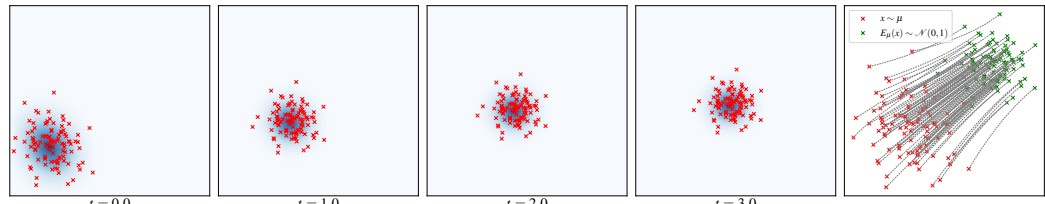

Figure 1: A toy example of our approach on a $2d$ multivariate normal distribution $\mu$. The first four plots visualize the diffusion process. The fifth plot demonstrates the trajectories of the probability flow ODE. In this case, the optimal transport map is known analytically and exactly coincides with the mapping $E_\mu$, introduced in Section 3.2. Note that the trajectories of the probability flow ODE are not straight lines even in this simple case.

is, in fact, the *gradient flow* of a certain functional in the *Wasserstein space* of probability measures. Moreover, locally, optimal transport maps between two 'consecutive' densities (i.e., separated by an infinitesimal time interval) along this trajectory are given precisely by the flow of the ODE (6). Thus, this trajectory is generally a sequence of infinitesimal optimal transports, which may not result in the 'global' optimal transport map. However, our experiments suggested that in the case when the target density is $\mathcal{N}(0, I)$ as in DDPMs, the map *is* practically indistinguishable from the optimal transport map. We start with a toy example and then proceed to more general cases.

## 3.1    Toy example

One of the few cases when both optimal transport and diffusion equations can be solved explicitly is the case of multivariate Gaussian distributions. For simplicity, in this example, we consider $\mu_0 \sim \mathcal{N}(a, I)$ with $a \in \mathbb{R}^n$ and $I \in \mathbb{R}^{n \times n}$ being the identity matrix. Rather than utilizing Equation (8) to obtain intermediate density values, we can do it directly from the transition formulas for SDE (5) found in Song et al. (2021b); Särkkä & Solin (2019); the corresponding probabilities then take the form:

$$\mu_t \sim \mathcal{N}(ae^{-t}, I). \tag{10}$$

The probability flow ODE takes the form $\frac{dx}{dt} = -ae^{-t}$. Solving this ODE amounts to simply integrating the function in the right-hand side, and in the limit $t \to \infty$ we obtain $E_{\mu_0}(x) = x - a$. Note that this is exactly the optimal transport map between $\mu_0$ and $\mathcal{N}(0, I)$.

## 3.2    Reminder on optimal transport

We now briefly recall the main definitions of the optimal transport theory. For a thorough review of optimal transport, we refer the reader to classical texts such as Thorpe (2018); Villani (2009); Ambrosio et al. (2005).

**Kantorovich and Monge formulation.**    Suppose we have two random variables with densities $\mu$ and $\nu$ supported on spaces $X$ and $Y$ respectively (in our work, we only consider distributions in Euclidean spaces). We consider the *optimal transport problem* between $\mu$ and $\nu$. There are several possible formulations of this task, namely the Kantorovich formulation and Monge formulation. Under quite general assumptions, these two formulations are equivalent. We will start from the Kantorovich formulation. In this formulation, we consider densities $\gamma \in \mathcal{P}(X \times Y)$ such that their marginal densities are $\mu$ and $\nu$ respectively. Such set is denoted by $\Pi(\mu, \nu)$ and its elements are termed *transport plans*. The Kantorovich formulation of the optimal transport problem is finding the *optimal* transport plan:

$$\int c(x, y)\gamma(x, y)dxdy \to \min_{\gamma \in \Pi(\mu,\nu)}, \tag{11}$$

where $c(x, y)$ is a cost function. Under quite general assumptions on $\mu, \nu$, there exists a unique solution to the Kantorovich problem. Rather than working with transport plans, it is more convenient to work with transport maps, which provide the means to actually move one distribution to another. In this formulation (termed the Monge formulation), our goal is to find a *transport map* $S : X \to Y$

such that

$$\int c(x, S(x))\mu(x)dx \to \inf_S,$$ (12)

over all $\mu$-measurable transport maps $S$ such that $\nu = S_\# \mu$. Here, $S_\# \mu$ is the push-forward measure, i.e., $(S_\# \mu)(A) = \mu(S^{-1}(A))$ for all measurable sets $A$.

Importantly, for *non-atomic* measures, these two definitions are equivalent (Villani, 2009). Given an optimal transport map, it can be straightforwardly converted to the optimal transport plan. Such a plan is deterministic, as we know precisely where to move each point. In this work, we consider the quadratic cost function, i.e., $c(x, y) = \|x - y\|^2$, as commonly done in the study of diffusion equations and gradient flows in Wasserstein spaces Villani (2009). In this case, the distance between distributions is denoted as $W_2(\cdot, \cdot)$ and is given by the square root of the optimal transport cost (11). In the appendix, we also discuss the Benamou-Brenier formulation of optimal transport.

### 3.3 MULTIVARIATE NORMAL DISTRIBUTION CASE

We start by showing theoretically that for arbitrary multivariate normal distributions, the DDPM encoder map coincides with the optimal transport map. This map is known analytically for normal distributions in the case of the quadratic cost function (Givens & Shortt, 1984).

**Theorem 3.1.** *Let $\mu_0 \sim \mathcal{N}(a(0), \Sigma(0))$ be a multivariate normal distribution. Then $E_{\mu_0}$ is the Monge optimal transport map between $\mu_0$ and $\mathcal{N}(0, I)$, i.e.,*

$$E_{\mu_0}(x) = \Sigma(0)^{-1/2}(x - a(0)).$$

See the appendix for the proof.

**Discussion.** We show that for multivariate normal distributions, the probability flow induced by the DDPM diffusion reduces to the optimal transport map in the limit. Despite the apparent simplicity of this case, the proof requires a non-trivial argument about the probability flow ODE. Thus, it is hard to attribute this result to a mere coincidence. We hypothesize that the statement of Theorem 3.1 is valid, perhaps, approximately, for arbitrary distributions (or at least a very large class of distributions), but defer more precise theoretical analysis of this statement to future work. For now, we thoroughly evaluate this hypothesis on a large class of synthetic and real distributions and show that it holds with high numerical precision. Very recently, in Lavenant & Santambrogio (2022) appeared a counterexample to the above statement with the two maps not being exactly equal (at a single point). However, our further analysis suggests that when given a finite number of samples in higher dimensions, the difference is still extremely low.

### 3.4 NUMERICAL EXPERIMENTS ON SYNTHETIC DATA

In this section, we numerically evaluate the aforementioned optimal transport map hypothesis. The main computational difficulty is the solution of the multidimensional Fokker-Planck equation (8). To maintain accuracy in traditional discretization-based numerical methods, the number of degrees of freedom of the approximation, i.e., the number of unknowns, grows exponentially as the dimensionality of the underlying state-space increases. In recent years, low-rank (low-parametric) tensor approximations have become especially popular for solving multidimensional problems in various fields of knowledge (Cichocki et al., 2016; 2017). Recently proposed approach (Chertkov & Oseledets, 2021) for the solution of the Fokker-Planck equation utilizing low-rank tensor train (TT) format (Oseledets, 2011), Chebyshev interpolation, splitting and spectral differentiation techniques, allowing us to consider sufficiently fine grids.

**Method.** We start by numerically solving the Fokker-Planck equation for an initial density $p_0(x) \equiv p(x, 0)$. According to approach (Chertkov & Oseledets, 2021), Equation (8) is discretized on a tensor-product Chebyshev grid with $N$ nodes for each dimension, and the density $p$ at each time step is represented as a $d$-dimensional tensor (array) $\mathcal{P}$. This tensor is approximated in the low-rank TT-format

$$\mathcal{P}[n_1, n_2, \ldots, n_d] \approx \sum_{r_1=1}^{R_1} \sum_{r_2=1}^{R_2} \cdots \sum_{r_{d-1}=1}^{R_{d-1}} \mathcal{G}_1[1, n_1, r_1]\mathcal{G}_2[r_1, n_2, r_2] \ldots \mathcal{G}_d[r_{d-1}, n_d, 1],$$ (13)

where $n_k = 1, 2, \ldots, N$ ($k = 1, 2, \ldots, d$) represent the multi-index, three-dimensional tensors $\mathcal{G}_k \in \mathbb{R}^{R_{k-1} \times N \times R_k}$ are named TT-cores, and integers $R_0, R_1, \ldots, R_d$ (with convention $R_0 = R_d = 1$) are named TT-ranks. Storage of the TT-cores $\mathcal{G}_1, \mathcal{G}_2, \ldots, \mathcal{G}_d$ requires less or equal than $d \times N \times \max_{1 \leq k \leq d} R_k^2$ memory cells instead of $N^d$ cells for the uncompressed tensor, and hence the TT-decomposition is free from the curse of dimensionality if the TT-ranks are bounded. Thus, at each time step $m = 1, 2, \ldots, \frac{t_{max}}{h}$, we obtain discrete values of the density $p(x, mh)$, represented in the compact TT-format as in Equation (13). In the appendix, we describe the algorithm in more details.

Next, we solve the probability flow ODE from Equation (9). We sample a set of points, termed $X_0$, from the original density $p_0(x)$ (since the initial density is presented in the TT-format, we use a specialized method for sampling from (Dolgov & Scheichl, 2019)). Then we numerically solve the probability flow ODE with the Runge-Kutta method (DeVries & Wolf, 1994) for each point (we calculate the density logarithm gradient using the spectral differentiation methods in the TT-format from (Chertkov & Oseledets, 2021)). The resulting set is denoted as $X_1$.

Given these two point clouds, $X_0$ and $X_1$, we numerically compute the optimal transport cost with the `Python Optimal Transport` (POT) library (Flamary et al., 2021). We then compute the transport cost for the map sending a point $x_i \in X_0$ to its counterpart $E_p(x_i) \in X_1$. Our hypothesis suggests that these two costs will be equal to a reasonable precision.

**Distributions.** We consider $d$-dimensional distributions specified by the following set of parameters: $a_1 \in \mathbb{R}^d, a_2 \in \mathbb{R}^d, Q_1 \in \mathbb{R}^{d \times d}, Q_2 \in \mathbb{R}^{d \times d}$, with $Q_1$ and $Q_2$ being symmetric positive definite. Given these parameters, we set the density $p(x)$ (with $x \in \mathbb{R}^d$) by the following formula

$$p(x) = \frac{1}{Z} \exp\left(-q_1(x) - q_2(x)\right),$$

$$q_1(x) = (x - a_1)^T Q_1 (x - a_1), \quad q_2(x) = \left((x - a_2)^2\right)^T Q_2 (x - a_2)^2,$$

where the square in $q_2$ is understood elementwise (so $q_2$ is a polynomial of degree 4) and $Z$ is a normalizing constant obtained by numerical integration. We then consider uniform mixtures of up to 5 of such randomly generated distributions, i.e.,

$$p(x) = \frac{1}{K} \sum_{i=1}^{K} p_i(x), \quad K \sim \text{Uniform}(5), \tag{14}$$

with $p_i$ being as above. Some examples for the 2-dimensional case (i.e., $d = 2$) are visualized at Figure 2. We see that the distributions are quite diverse and lack any symmetry.

We use the distribution from Equation (14) for the 2 and 3-dimensional case. If $d > 3$, the number of parameters in the TT-representation of this distribution turns out to be too large, so we use a density of a simpler form

$$\hat{p}(x) = \frac{1}{Z} q(x) \cdot (2\pi)^{-\frac{d}{2}} \cdot exp\left(-\frac{||x||^2}{2}\right), \tag{15}$$

where $Z$ is a normalizing constant obtained by numerical integration in the TT-format and $q(x)$ is a random function in the TT-format with positive TT-cores (each element of the TT-cores was sampled from $\mathcal{U}(0, 1)$) and of rank 2. Since the Gaussian density has rank 1 with variables being completely separated, the above formula provides us with a low-rank fast decaying density suitable for experiments.

**Experiment and results.** We consider three dimension numbers $d = 2, 3, 7$. For the case $d = 2$ and $d = 3$ we use the density from Equation (14) and for the case $d = 7$ we use the density from Equation (15). As the spatial domain, we consider the square $[-8, 8]^d$, and densities were scaled beforehand to decay sufficiently fast towards the boundary. For the time dimension, we consider the range $[0, 5]$. We construct 100 random densities for each $d$ and compute two transport costs as described above. Some examples of the obtained encoder maps for the 2-dimensional case are given at Figure 3. The trajectories of the probability flow ODE are highly nonlinear, despite providing the optimal transport map in the limit.

The obtained results are presented at Table 1. Here, the error is defined as

$$\varepsilon_{rel}(p) = \frac{\text{Cost}(E_p) - \text{Cost}(OT)}{\text{Cost}(OT)},$$

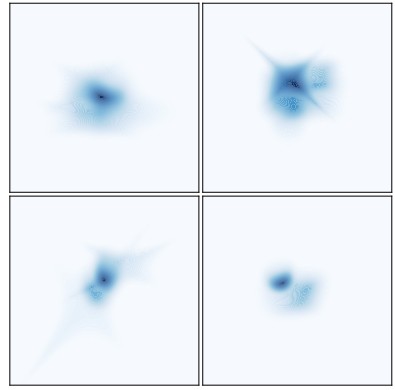

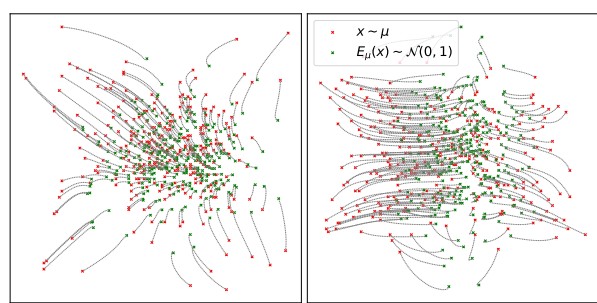

Figure 2: An example of 2-dimensional distributions considered for numerical comparison of the Monge optimal transport map against the DDPM encoder map.

Figure 3: An example of the trajectories of the probability flow ODE and the limiting encoder map for two 2-dimensional distributions studied in Section 3.4.

Table 1: Numerical results for synthetic data of different dimensions.

| Dimensionality $d$ | Spatial grid size | Temporal grid size | Maximum error $\varepsilon_{rel}(p)$ |
|---|---|---|---|
| 2 | 250 | 250 | $5.7 \cdot 10^{-15}$ |
| 3 | 100 | 100 | $2.2 \cdot 10^{-15}$ |
| 7 | 50 | 50 | $2.1 \cdot 10^{-15}$ |

and in the table we report the obtained maximum value of the error for 100 random densities. For all three considered settings, we have a very high (up to machine precision) accuracy, and we conclude that this experiment fully supported our hypothesis. Note that these calculations were carried out on a regular laptop and run for one random density took about 100 seconds on average for each model problem. We have additionally run this experiment for the counterexample provided in Lavenant & Santambrogio (2022), again obtaining a machine precision level difference, which is not surprising since the difference between two maps is shown to hold only at $(0,0)$. This suggests that constructing a 'numerical' counterexample is a nontrivial task. The code is available in the supplementary material.

We now attempt to verify it on a more qualitative level for high-dimensional distributions, where solving the Fokker-Planck equation directly is infeasible. Namely, we consider DDPMs trained on high-resolution image data.

## 3.5 EXPERIMENTS OF IMAGE DATASETS

Our experiments in this section are organized as follows. Suppose that we are given two datasets and train a DDPM for each dataset. Then, by the hypothesis, each DDPM encoder performs optimal transport from respective distributions to $\mathcal{N}(0, I)$. Since the optimal transport in our setting is performed in the $L_2$ sense, for images, this will be reflected by similarity in the pixel space, i.e., high-level texture similarity. We can perform the following experiment to see if this indeed holds. Take a latent code from $\mathcal{N}(0, I)$, and compute a corresponding image from each distribution respectively by reversing the encoder map. We can now compare if these images share texture and a high level semantic structure. Indeed, if each of those images is $L_2$-close 'on average' to the same latent code, we expect them to also be similar on the pixel level. Additionally, on datasets of relatively small sizes, we again can compare the directly computed OT map in the pixel space with the DDPM encoder map.

**Datasets.** We consider the AFHQ animal dataset (Choi et al., 2020). It consists of 15000 images split into 3 categories: cat, dog, and wild. This is a common benchmark for image-to-image methods. We also verify our theory on the FFHQ dataset of 70.000 human faces (Karras et al., 2019) and the

MetFaces dataset (Karras et al., 2020) consisting of 1000 human portraits. Finally, we consider a conditional DDPM on the ImageNet dataset (Deng et al., 2009). By changing the conditioning label, we can control what distribution is being produced, and the argument above still holds.

**Models.** We consider guided-diffusion, state-of-the-art DDPMs (Dhariwal & Nichol, 2021). We use the official implementation available at github [1]. For each of the datasets we train a separate DDPM model with the same config as utilized for the LSUN datasets in Dhariwal & Nichol (2021) (with dropout); we use default 1000 timesteps for sampling. The AFHQ models were trained for $3 \cdot 10^5$ steps, the FFHQ model was trained for $10^6$ steps; for the MetFaces model, we finetune the FFHQ checkpoint for $25 \cdot 10^3$ steps similar to Choi et al. (2021). All models were trained on $256 \times 256$ resolution. For the ImageNet experiment, we utilize the 256 resolution checkpoint for the conditional model available at the GitHub link above.

**Algorithm.** We follow the aforementioned experimental setup: we sample a number of latent codes from $\mathcal{N}(0, I)$ in the pixel space ($\mathbb{R}^{256 \times 256 \times 3}$) and decode them by different DDPMs with the DDIM algorithm. For the ImageNet, we have a single DDPM model but vary the conditioning label.

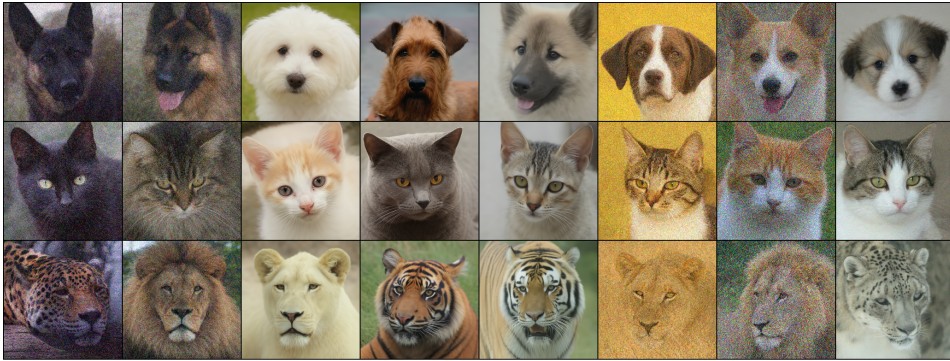

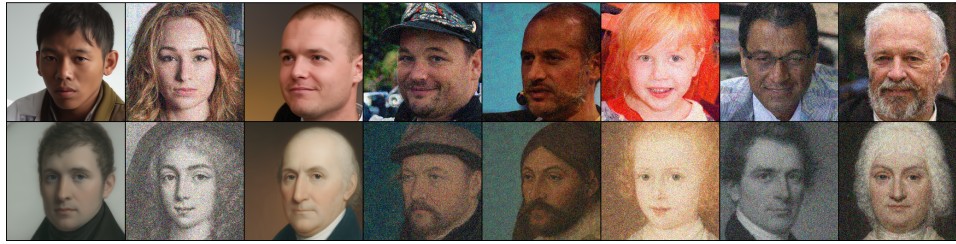

Figure 4: Examples of synthetic samples produced with DDIM sampling from the same latent codes. (*Top*) Three independent DDPMs trained on AFHQ *Dog/Cat/Wild*. Each row translates into each row in most cases preserving high-level semantics such as pose and texture, thus supporting the claim of our hypothesis. (*Bottom*) Two DDPMs trained on FFHQ/MetFaces; high-level features such as gender and texture/color are transferred. 'Noisy' samples seem also to be shared across models.

**Qualitative results.** We fix a number of latent codes and produce images by all the AFHQ models. The obtained samples are visualized on Figure 4 (top). We observe that samples from different rows indeed share high-level features such as texture and pose. Similarly, we visualize samples from FFHQ and MetFaces on Figure 4 (bottom). We note that samples are aligned in this case as well; notably, in most cases, pose, gender, and various features such as hairstyle and mustache are transferred. Interestingly, artifacts of models such as slight noisiness also seem to be shared.

Results obtained with the ImageNet model are provided at Figure 5. We selected a number of diverse classes, namely, *rooster*, *pizza*, *crab*, *corgi*. We sampled a number of latent codes and produced images by conditioning on the respective label. We see that in this case, high-level textures are transferred as well, sometimes in quite amusing ways (e.g., rooster/pizza rows). As a sanity check, we performed a similar experiment with the conditional BigGAN model (Brock et al., 2019) pretrained

---

[1]https://github.com/openai/guided-diffusion

on ImageNet. We did not observe similar behavior, suggesting that the texture alignment is an artifact specific to DDPM and not happening simply due to the conditioning mechanism.

**Numerical results.** We now apply the evaluation scheme of Section 3.4 to the AFHQ dataset. We take the validation subset of each part of the dataset (*Cat*/*Dog*/*Wild*) containing 500 images and invert them with DDIM. We then compute the optimal transport matrix between the images and their respective codes (understood as vectors in $\mathbb{R}^{256 \times 256 \times 3}$) with POT. We then compared the obtained OT cost with the cost for the ODE induced map. In all three cases, they **matched up to a machine precision**. Given the high dimensionality of the pixel space and a low number of samples, the exact coincidence with the hypothesis is not surprising, since for the majority of synthetic experiments, there was an exact match. Intuitively, this should become easier in high dimensions as we may trade some inaccuracies in the model with there being 'more space'.

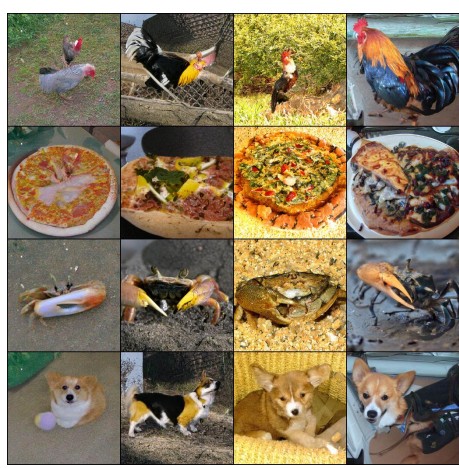

To conclude, experiments on real image datasets support our hypothesis and provide interesting examples of how latent codes can be utilized to transfer images from one distribution to another via a 'proxy' normal distribution.

Figure 5: Examples of synthetic samples produced with DDIM sampling from same latent codes for different classes with a conditional ImageNet model. We observe that samples share texture and pixel-based similarity.

## 4 RELATED WORK

OT is a classical area of mathematics tightly related to diffusion processes and PDEs. With hundreds of papers on this topic, we can not provide a thorough review of all the literature and refer the reader to great texts such as Villani (2009); Ambrosio et al. (2005); Thorpe (2018). Computational approaches to OT have been extensively studied by the machine learning communities (Cuturi, 2013; Altschuler et al., 2017; Lin et al., 2019). We also note the work (Onken et al., 2021) in which the relationship between OT and normalizing flows was considered, as well as the work (Mokrov et al., 2021), where a method based on the Wasserstein gradient flows was proposed for solution of the Fokker-Planck equation. A recently introduced class of score-based models termed diffusion Schrödinger bridges (Chen et al., 2016; De Bortoli et al., 2021; Gushchin et al., 2022) is tightly related to the entropy-regularized OT problem, providing a continuous analog of the Sinkhorn algorithm. In our paper, we show a connection between DDIM and (non-regularized) OT formulations, not covered by the Schrödinger bridge theory. As mentioned in Section 3.2, a similar question was studied in Lavenant & Santambrogio (2022) providing a theoretical counterexample, while our analysis is mostly numerical and is motivated by the practical goal of understanding the DDPM encoder.

## 5 CONCLUSION

In this work, we provided theoretical and experimental evidence that due to the nature of DDPMs, the encoder map turns out to be the optimal transport map. We believe that this result will be interesting to both optimal transport and diffusion models communities. We hope that it will inspire other researchers to obtain general proof of this hypothesis. One observation about the obtained results is that the quadratic cost in the pixel space seems suboptimal when working with visual data. Perhaps, a more elaborate diffusion process leading to feature-based cost may be constructed. We leave this analysis to future work.

## ACKNOWLEDGMENTS

The work was supported by the Analytical center under the RF Government (subsidy agreement 000000D730321P5Q0002, Grant No. 70-2021-00145 02.11.2021).

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

## A  BENAMOU-BRENIER FORMULATION OF OPTIMAL TRANSPORT

It will be instructive to consider another formulation of the optimal transport, originating in spirit from the fluid dynamics (Benamou & Brenier, 2000; Villani, 2003). Assume that at $t = 0$ we are given a set of 'particles' from the density $\rho_0$ which move in such a way that at $t = 1$ their state is described by the density $\rho_1$. Moreover, these particles move in such that they perform the least amount of work. Formally, they minimize the following *action*:

$$A = \int_0^1 \left( \sum_x |\dot{x}(t)|^2 \right) dt,$$

with $x$ varying in the set of particles. In the continuous limit, we obtain the following formulation (with $v_t$ being the velocity field):

$$\inf_{\rho,v} \left\{ \int_0^1 \int \rho_t(x) |v_t(x)|^2 \, dxdt; \ \frac{\partial \rho_t}{\partial t} + \nabla \cdot (\rho_t v_t) = 0 \right\} \quad (16)$$

where the infimum is taken over all time-dependent probability densities $\{\rho_t\}_{t=0}^1$ which agree with $\rho_0$ and $\rho_1$ at respective times $t = 0$ and $t = 1$, and overall time-dependent velocity fields $\{v_t\}_{t=0}^1$ which convect $\rho_t$, as expressed by the continuity equation in the right-hand side of Equation (16).

It can be shown that the infimum in Equation (16) is given by $W_2(\rho_0, \rho_1)^2$. Moreover, given an optimal transport map the corresponding admissible trajectory for the Benamou-Brenier formulation can be easily constructed. Importantly, trajectories generated by this vector field are *straight lines*, which does not hold for the probability flow ODE (see, e.g., Figure 1). There is no contradiction here: we do not claim that the probability flow ODE is the solution of the Benamou-Brenier problem, but only that the flow map induced by this ODE (in the limit $t \to \infty$) solves the Monge problem. So, in general, the vector field solving (16) and the vector field of the probability flow ODE will be different.

## B  PROOF OF THEOREM 3.1

The proof of Theorem 3.1 is based on the fact that in the case of $\mu_0$ being normal distribution, all the intermediate densities $\mu_t$ are normal as well, i.e, $\mu_t \sim \mathcal{N}(a(t), \Sigma(t))$. Moreover, we can easily obtain ODEs describing how their means and covariances transform (Song et al., 2021b; Särkkä & Solin, 2019). These ODEs take the following form:

$$\frac{d\Sigma(t)}{dt} = 2(I - \Sigma(t)); \quad \frac{da(t)}{dt} = -a(t),$$

with the solutions

$$\Sigma(t) = I + e^{-2t}(\Sigma(0) - I); \quad a(t) = e^{-t}a(0).$$

Let us start with the homogeneous case $a(0) = 0$, so $a(t) \equiv 0$ (we provide the proof of the general case below). The probability flow ODE then takes the following form:

$$\frac{dx}{dt} = -(I - \Sigma^{-1}(t))x. \quad (17)$$

We start by noticing that all the matrices $\Sigma(t)$ commute with each other. We then can write the solution of Equation (20) as:

$$x(t) = \exp\left( -\int_0^t I - \Sigma^{-1}(\tau)d\tau \right) x(0),$$

with $\exp$ being the matrix exponential. Let us consider the eigendecomposition of $\Sigma(0)$, i.e., assume that $\Sigma(0) = U^T \Lambda U$ with $\Lambda = \mathrm{diag}(\lambda_1, \ldots, \lambda_d)$ being a diagonal matrix of positive eigenvalues and $U^T U = I$. We obtain that

$$x(t) = U^T \mathrm{diag}(f(\lambda_1, t), f(\lambda_2, t), \ldots f(\lambda_d, t)) U x(0), \quad (18)$$

where $f(\lambda, t)$ is a function defined as

$$f(\lambda, t) = \exp\left(-\int_0^t 1 - (1 + e^{-2\tau}(\lambda - 1))^{-1} d\tau\right),$$

where now all the functions operate on real numbers. Direct integration shows that:

$$
\begin{aligned}
f(\lambda, t) &= \exp\left(-t + \frac{1}{2}\log\left(\lambda - 1 + e^{2t}\right) - \frac{1}{2}\log\lambda\right) \\
&= \sqrt{\frac{\lambda - 1 + e^{2t}}{\lambda e^{2t}}} = \sqrt{\frac{e^{-2t}(\lambda - 1) + 1}{\lambda}}.
\end{aligned}
\tag{19}
$$

We see that $\lim_{t\to\infty} f(\lambda, t) = \lambda^{-1/2}$. Then, by definition of the matrix square root, $E_{\mu_0}(x) = \Sigma(0)^{-1/2}x$. This is exactly the optimal transport map between $\mu_0$ and $\mathcal{N}(0, I)$.

For a general case, we obtain the following ODE.

$$\frac{dx}{dt} = -(I - \Sigma^{-1}(t))x - \Sigma^{-1}(t)a(t). \tag{20}$$

This is an inhomogeneous ODE. We can obtain its solution via the following standard approach. Let $Z(t)$ be the fundamental solution of the homogeneous part (see Equation (18)), i.e,

$$Z(t) = U^T \mathrm{diag}(f(\lambda_1, t), f(\lambda_2, t), \dots f(\lambda_d, t))U,$$

with

$$f(\lambda, t) = \sqrt{\frac{e^{-2t}(\lambda - 1) + 1}{\lambda}}.$$

We find that a particular solution (with 0 initial condition) of the inhomogeneous problem is given by:

$$\bar{x}(t) = -Z(t)\int_0^t Z^{-1}(\tau)\Sigma^{-1}(\tau)a(\tau)d\tau.$$

Recall that

$$\Sigma(t) = I + e^{-2t}(\Sigma(0) - I); \quad a(t) = e^{-t}a(0).$$

We obtain:

$$\bar{x}(t) = \left(-Z(t)\int_0^t Z^{-1}(\tau)\Sigma^{-1}(\tau)e^{-\tau}d\tau\right)a(0).$$

Again, it suffices to work on the level of the spectrum. I.e,

$$\bar{x}(t) = U^T \mathrm{diag}(g(\lambda_1, t), g(\lambda_2, t), \dots, g(\lambda_d, t))Ua(0),$$

with

$$
\begin{aligned}
g(\lambda, t) &= -f(\lambda, t)\int_0^t \frac{e^{-\tau}}{(1 + e^{-2\tau}(\lambda - 1))f(\lambda, \tau)}d\tau \\
&= -\frac{f(\lambda, t)}{\lambda}\int_0^t \frac{e^{-\tau}}{f^3(\lambda, \tau)}d\tau \\
&= -\sqrt{\lambda}f(\lambda, t)\int_0^t \frac{e^{-\tau}}{(e^{-2\tau}(\lambda - 1) + 1)^{3/2}}d\tau.
\end{aligned}
\tag{21}
$$

Direct integration shows that:

$$\lim_{t\to\infty} g(\lambda, t) = -\frac{1}{\sqrt{\lambda}}.$$

Thus, we obtain that

$$\lim_{t\to\infty} \bar{x}(t) = -\Sigma(0)^{-1/2}a(0).$$

By combining this piece with the limit of general solution obtained above, we find that

$$E_{\mu_0}(x) = \Sigma(0)^{-1/2}x - \Sigma(0)^{-1/2}a(0) = \Sigma^{-1/2}(0)(x - a(0)),$$

which is exactly the desired optimal transport map between $\mu_0$ and $\mathcal{N}(0, 1)$ (Dowson & Landau, 1982; Olkin & Pukelsheim, 1982). This expression is also familiar from the commonly used Frechét Inception Distance metric, where the OT distance is used to evaluate generative models (Heusel et al., 2017).

# C  TENSOR-TRAIN SOLVER FOR THE FOKKER-PLANCK EQUATION

Let us briefly summarize the algorithm from (Chertkov & Oseledets, 2021) for solving the Fokker-Planck equation. If we introduce diffusion and convection operators from Equation (8)

$$\widehat{V}p(x,t) \equiv \nabla_x^2 p(x,t), \quad \widehat{W}p(x,t) \equiv \nabla_x\left(x\,p(x,t)\right),$$

then on each time step $m$ $(m = 1, 2, \ldots, M)$ the standard second order operator splitting technique (Glowinski et al., 2017) may be formulated

$$p_{m+1} = e^{h\left(\widehat{V}+\widehat{W}\right)}p_m \approx e^{\frac{h}{2}\widehat{V}}e^{h\widehat{W}}e^{\frac{h}{2}\widehat{V}}p_m,$$

where $h$ is the size of the temporal grid and $p_m(x) = p(x, mh)$ is a solution on the $m$-th time step. It is equivalent to the sequential solution of the following equations

$$\frac{\partial p_D^{(1)}}{\partial t} = \nabla_x^2 p_D^{(1)}, \quad p_D^{(1)}(\,\cdot\,,t_m) = \rho_m(\,\cdot\,), \tag{22}$$

$$\frac{\partial p_C}{\partial t} = \nabla_x(xp_C), \quad p_C(\,\cdot\,,t_m) = p_C^{(1)}(\,\cdot\,,t_m + \frac{h}{2}), \tag{23}$$

$$\frac{\partial p_D^{(2)}}{\partial t} = \nabla_x^2 p_D^{(2)}, \quad p_D^{(2)}(\,\cdot\,,t_m) = p_C(\,\cdot\,,t_m + h), \tag{24}$$

with the final approximation of the solution $\rho_{m+1}(\,\cdot\,) = p_D^{(2)}(\,\cdot\,,t_m + \frac{h}{2})$.

To solve the diffusion part (see Equations (22) and (24)), we discretize the Laplace operator using the second order Chebyshev differential matrices (Trefethen, 2000), and convolve the matrix exponential with the TT-cores of $p_m$. The convection part (see Equation (23)) is solved by efficient interpolation in the TT-format with the TT-cross method (Oseledets & Tyrtyshnikov, 2010). Finally, at each time step $m$, we obtain discrete values of the density $p(x, mh)$, represented in the low rank TT-format as in Equation (13). Then we can solve the probability flow ODE, using these tensors, as described in the main text.

