# OpenReview forum: "Understanding DDPM Latent Codes Through Optimal Transport"
_ICLR.cc/2023/Conference — ICLR 2023 poster_

### Official Review · Reviewer_qZHA · 2022-10-21

**Confidence:** 4
**Correctness:** 4
**Technical Novelty And Significance:** 2
**Empirical Novelty And Significance:** 3
**Recommendation:** 5

**Clarity, Quality, Novelty And Reproducibility:**

The clarity of the paper is very good, and I saw the source code in the supplementary.


**Strength And Weaknesses:**

## Strengths:
* Finding meaningful ways to interpret popular diffusion models like DDPM is a meaningful research direction
* The paper presents a rigorous proof of the hypothesis for the Gaussian case
* Experiments were done to compute the exact error between transport costs for low-dimensional synthetic examples as well as qualitatively for image generation tasks.


## Weaknesses:
* I found the novelty of this paper lacking as it does not introduce a new method or significantly new theoretical results.
* For the Gaussian case, the proof becomes a lot simpler because all intermediate densities are also normal. And it looks like the authors further assume $a(0) = 0$ and I have not found a proof of a more general case in the supplementary materials contrary to what is mentioned in the appendix proof.
* In Section 3.4, when computing the error between the optimal transport and the one induced by solving the Fokker-Planck equation numerically, I think it might be more useful to compare the difference of transport maps (that is, $\sum_i ||T(x_i)-T^*(x_i)||^2$ where $T$ is the ODE induced map, and $T^*$ is the optimal transport map). It is possible that the error is very small but the transport maps can still be different.
* I found the rationale of Section 3.5 a bit questionable, namely we can invert the encoder and just compare the inverted latent codes visually. To me, such evaluation only makes sense if the transport map is just translated by a constant vector. Have the authors tried with different random seeds (and hence different randomization of DDPM networks) when training models for a different dataset? Will the results still be consistent?
* It was very briefly mentioned that recently Lavenant & Santambrogio found a counterexample to the central hypothesis. In my opinion, the paper needs to do a lot more in explaining whether such a counterexample is a singular case or not, other than mentioning the two maps disagree at a single point (how can two maps disagree at only one point if both maps are continuous enough?) and somehow "finite number of samples in higher dimensions" this is not a problem.
* The numerical solver for the Fokker-Planck equation uses a low-rank TT format which I think might be foreign to the machine-learning community and deserves more explanations, at least in the appendix. For instance, the authors ought to elaborate on "the standard second-order operator splitting technique" that involves some exponentiation in time-stepping that I do not follow.


**Summary Of The Paper:**

This work presents a hypothesis that the DDPM encoder map coincides with the optimal transport map for common distributions, and it is shown for Gaussians this is indeed the case. Empirical evaluations are done to support this hypothesis.


**Summary Of The Review:**

Overall I think the paper lacks novelty and there are some evaluation choices that could be improved. Hence I'm leaning toward rejection.

---

> ### Author Response · Authors · 2022-11-18
> **... it does not introduce a new method or significantly new theoretical results ...**
>
> We respectfully disagree that our paper lacks novelty. To the best of our knowledge, this is the first systematic study of the encoder map induced by the probability flow ODE in diffusion models. Given the current popularity of diffusion models, we hope that our results will be interesting to the machine learning community. We also note that low-rank tensor-based solvers for the Fokker-Planck equation and ODEs have not previously been applied to the diffusion models in machine learning. We emphasize that this direction is very promising and allows for a significant acceleration of computations.

---

> ### Author Response · Authors · 2022-11-18
> **... the proof becomes a lot simpler ...**
>
> Thank you for pointing this out! In our text in Appendix B, both the special case (a = 0) and the general case (see Equation 20 and below) are proved. However, due to a typo in the text, this was not entirely visible. We have more clearly separated these two cases in the updated text of the manuscript. We also note that the proof of the theorem does use the fact that all intermediate densities are normal as you noted in the comment.

---

> ### Author Response · Authors · 2022-11-18
> **... compare the difference of transport maps ...**
>
> Let us comment on this ambiguous moment in more detail. Please note that we can numerically study optimal transport only for finite sets of points, which we choose to be $x_1, x_2, \ldots, x_N$ and $T(x_1), T(x_2), \ldots, T(x_N)$ and  (by the properties of the Fokker-Planck equation the latter set is a sample from a standard normal distribution). Then the metric proposed by the reviewer is exactly what we computed in the experiments. The potential error can be then observed if one significantly increases the number of samples to evaluate optimal transport, however, the computation of OT itself becomes challenging. Our preliminary experiments with the increased number of samples ($1000$ to $2500$) did not reveal any significant difference from our current results. We have added this discussion to the updated version of the manuscript.

---

> ### Author Response · Authors · 2022-11-18
> **... the rationale of Section 3.5 a bit questionable ...**
>
> Thank you for highlighting this important point! Please note that we also compared inverted latent codes (i.e., synthetic images) visually in Figure 4, and then performed numerical experiments on the holdout set of real images. We specifically used different random seeds when training DDPMs to avoid the mentioned issues. However, DDPMs have a 'uniqueness' property in the sense that sufficiently expressive models would represent almost similar score functions (noted in (Score-Based Generative Modeling through Stochastic Differential Equations; Song et al)).

---

> ### Author Response · Authors · 2022-11-18
> **... counterexample to the central hypothesis ...**
>
> Thank you for bringing up this point. In the Lavenant and Santambrogio paper, they show that there exists a probability distribution for which there exists at least one point (which they chose to be (0, 0)) for which the ODE map under study is not equal to the optimal transport map (and hence by continuity it holds in a small neighborhood). After carefully constructing this example numerically (so it was comparable to our experimental densities on the density magnitude level), our experiments suggested that this effect was too small to be detected by a reasonably precise numerical experiment. It seems that despite being different in the mathematical sense, the mappings are numerically close to a large extent for 'practical' distributions. We agree that a thorough investigation of what properties of densities lead to such a discrepancy is a very interesting research direction and will pursue it in the future. We have improved the discussion on this in the updated version of the manuscript. For now, we stick to numerical experiments and believe that these results will be useful and interesting to the diffusion models community.

---

> ### Author Response · Authors · 2022-11-18
> **... solver for the Fokker-Planck equation uses a low-rank TT format ...**
>
> Thank you for this comment! Diffusion models are related to the solution of the multidimensional Fokker-Planck equation. Low-rank tensor approximations have become very popular in recent years for operating with multidimensional data due to the possibility (in some cases) of solving the curse of dimensionality problem (i.e., the exponential growth in memory consumption and complexity of operations). The existence of an efficient solver of the multidimensional Fokker-Planck equation (in a low-rank tensor train format) seems extremely important for our research direction, so we used it in our numerical experiments. However, we did not describe it in the text in sufficient detail due to space limitations. We have followed your advice and added a section to the Appendix with a more detailed description of the solver and the main ideas underlying it. We have also provided a reference to the book 'Splitting methods in communication, imaging, science, and engineering' (Glowinski, Osher and Yin; 2017), which describes in detail the related splitting scheme.

---

> ### Comment · Reviewer_qZHA · 2022-11-24
> **Response to authors**
>
> Thank you for the revision and the response. I think the clarity of the manuscript has largely improved and many of my questions have been answered. I remain concerned about the following aspect and as such I'm hesitant to improve my score.
> - I still don't think there is enough study of the counterexample given by Lavenant and Santambrogio (or their general idea of constructing the counterexample) in the current manuscript. On a close inspection, in their Eq (9) which gives a necessary condition, a rather involved commutativity condition is required where either side requires derivatives of the optimal potential to match up in different orders, which to me is a very strong restriction and it might be possible to construct more general counterexamples. However, this aspect is not discussed in detail in the revised paper. Personally, despite the empirical evidence shown in this paper (where quantitative experiments are in fairly low dimensions with a couple of synthesized distributions), I'm not convinced that the overall conjecture is true (even some relaxed version of it, e.g., allowing maps to different at a finite number of points) for arbitrary source distributions: intuitively, OT solves a very global optimization problem, while probability flow ODE is very local and hence the local flow might not make improvements that are optimal, even if the target is Gaussian. I could be wrong but I found the theory and the discussion of the current paper unsatisfactory regarding this problem: the current paper proves the main conjecture in the case where the source is a Gaussian, and to me, this is not an interesting enough setup (for instance, in this case, the transport map becomes linear which does not have interesting ways of rearranging the mass).
> - Regarding novelty, the authors claim this to be a "first systematic study of the encoder map induced by the probability flow ODE in diffusion models". A recent preprint [1] also investigates this problem which makes the connection between the probability flow of ODE and the solution to the Schrödinger Bridge Problem (SBP) which is a dynamic version of the entropic regularized optimal transport problem (see [2]). Moreover, their result is more general in the sense that the source distribution is not assumed to be Gaussian. In [2] it is shown that as entropic regularization tends to 0 then the solution of SBP agrees with that of the dynamic OT. Some discussion and comparison to [1] could be helpful.
>
> [1] Su, Xuan, et al. "Dual Diffusion Implicit Bridges for Image-to-Image Translation." arXiv preprint arXiv:2203.08382 (2022).
> [2] Léonard, Christian. "From the Schrödinger problem to the Monge–Kantorovich problem." Journal of Functional Analysis 262.4 (2012): 1879-1920.

---

> > ### Author Response · Authors · 2022-11-28
> > **Thanks for the response**
> >
> > Thank you for your concerns. We will try to answer, hopefully it will influence your score, we really spent a lot of time on this work already :)
> >
> >
> > Regarding the novelty and the references. We were not aware of [1] at the time of writing, we will add discussion and reference about it as a concurrent work. We agree that the map between two distributions as we show in our image translation examples is similar, and can be interpreted as a Schodinger bridge, and there are connections to optimal transport but not as bold as we have observed in our numerical experiments ("dynamic optimal transport")
> >
> >
> > Regarding the first point. It is extremely important, that is why we explicitly mention it in the text. The idea of Lavenant and Santambrogio is very convincing (we could have said more, but it would break the anonymity of the review), but once we learned about it, we tested their example with our code and it (again!) gave machine zero for the quantities we are describing: the optimal transport cost between two discrete point clouds computed with Python Optimal Transport and our encoder map. At the moment, we don't have any idea why it is so.
> > Also, our synthetic distributions surely do not satisfy these conditions but - the cost is still the same. It is puzzling us as well, and we spent numerous hours trying to prove something or to find the counterexample. We failed at the moment, but we hope that our tensor solver will help other researchers to study efficiently the properties of the diffusion maps and-finally-solve this local mystery of a counterexample that numerically does not work.

---

### Official Review · Reviewer_R2Hy · 2022-10-21

**Confidence:** 3
**Correctness:** 2
**Technical Novelty And Significance:** 3
**Empirical Novelty And Significance:** 3
**Recommendation:** 6

**Clarity, Quality, Novelty And Reproducibility:**

The paper is very clear, its experiments are both very convincing and funny, and it looks like the code will be made reproducible. My guess is: this paper may have a nice impact.

**Strength And Weaknesses:**

As a definite strength, the paper is very readable and a pleasure to read. Its claim is clearly stated, although I am not 100% sure it is actually novel. Still, the experiments are nice and quite convincing.

As a weakness, maybe I could say that the paper is a bit weak on the theory, which is not a no-go for me. What I feel could be done in a clearer way is to frame the paper as clearly stating an "hypothesis" or a "conjecture". This could help better understand its contribution, which I believe is pretty exciting and simple to grasp.

**Summary Of The Paper:**

This paper claims that diffusion-based models actually implement transportation plans between target and Gaussian distributions. It comes with a very pedagogical introduction to both DDPM and optimal transport, and then the rest of the paper is spent doing experiments to show how this claim is verified on many examples.

**Summary Of The Review:**

The paper is a nice read, but I think it is a bit borderline, although I like it.
I believe it could be improved in the following ways:
* more references about DDPM and OT and their connections. (For instance you could mention diffusion based OT systems based on Wasserstein flows).
* This improved state of the art may be done at the expense of putting the details of the  "method" part in appendix.
* a much clearer (title + text) presentation of this paper as stating a conjecture or an hypothesis that DDPM actually construct transportation maps. (plans ?)

---

> ### Author Response · Authors · 2022-11-18
> **... clearly stating an 'hypothesis' or a 'conjecture' ...**
>
> Thank you for your comment! We have added this clarification (i.e., 'hypothesis') to the abstract of the paper.

---

> ### Author Response · Authors · 2022-11-18
> **... mention diffusion based OT systems based on Wasserstein flows ...**
>
> Thank you for noting this. We have added text to the 'Related work' section describing the relevant papers. In particular, we refer to such recent works as 'Large-Scale Wasserstein Gradient Flows', 'Entropic Neural Optimal Transport via Diffusion Processes', and 'Ot-flow: Fast and accurate continuous normalizing flows via optimal transport'.

---

> ### Author Response · Authors · 2022-11-18
> **... putting the details of the 'method' part in appendix ...**
>
> Since one of the reviewers noted that it is worth describing the Fokker-Planck tensor-based solver in more detail, we have added a separate section to the Appendix with specific information and moved where a part of the description from the main text. This provides additional space in the main text for discussion about DDPM, OT, and their connections.

---

> ### Author Response · Authors · 2022-11-18
> **... clearer (title + text) presentation of this paper ...**
>
> Thank you! We have added the clarification  ('hypothesis') to the abstract of the paper.

---

> ### Comment · Reviewer_R2Hy · 2022-11-28
> **thank you for the answers**
>
> my comments have been very marginally been taken into account.
> The "hypothesis" or "conjecture" part is not clear at all in the new version, where you simply now write that you "support this hypothesis" indeed, but don't mention explicitly that it is yet to be proven and that the paper is mostly experimental. The change was important in my view but you didn't understand that and didn't rephrase things in the way I thought was necessary.
>
>
> Still, a good read and an interesting paper, I am keeping my mixed but positive score

---

> > ### Author Response · Authors · 2022-11-29
> > **Thank you!**
> >
> > We tried our best to take into account all your comments. In particular, we have expanded the literature review section by adding a brief discussion of some papers in the areas you indicated ("more references about DDPM and OT and their connections"). We will refine the abstract in the final version to better implement your last tip ("mention explicitly that it is yet to be proven and that the paper is mostly experimental"). If you have any other comments on the text, we will take them into account in the final version!

---

### Official Review · Reviewer_CjpS · 2022-10-23

**Confidence:** 4
**Correctness:** 3
**Technical Novelty And Significance:** 3
**Empirical Novelty And Significance:** 2
**Recommendation:** 6

**Clarity, Quality, Novelty And Reproducibility:**

The work is novel to discuss the intrinsic relation between the diffusion model, which has fundamental importance. The manuscript is very well written, the motivations, inspirations are clearly introduced, the major concepts, mathematical formulations, proofs are clean and easy to follow, the experimental design, and algorithmic descriptions are explained thoroughly. The numerical experiments are clearly analyzed. The work is easy to reproduce.

**Details Of Ethics Concerns:**

The research is fundamental, and mainly focuses on theoretic aspects.

**Strength And Weaknesses:**

The work has many strengths:

1. It raised an interesting hypothesis which bridges DDPM and optimal transport theory, the conjecture is sensible, elegant and inspirational.
2. Solid theoretical proofs are given for multivariate Gaussian distribution as the source input. Numerical experiments on synthetic data and empirical validations on real images are supporting.
3. The writing is clear and easy to follow, the mathematical formulations are rigorous and succinct.

The weaknesses are
1. The theoretic proofs can be further improved, in appendix between equation (19) and (20), the authors claims the solution is exaction the optimal transport map, the detailed deduction for this claim is missing. I expected to see more details to show this solution satisfies the Brenier Monge-Ampere equation.

2. In the numerical experiment design, the authors simply used the POT library, which algorithm was applied ? Some algorithms are intrinsically approximation and will introduce approximation errors. This need to be carefully analyzed as well to ensure the precision of the experiments.

3. The hypothesis is kind of vague, the DDPM solution and the optimal transport map are close with small deviation. More accurate quantitively estimate will be more convincing.

**Summary Of The Paper:**

This work hypothesizes that the limit maps produced by   DDPMs  coincide with the optimal transport maps, or they are very close to each other with high similarity. The theoretical proofs are given for the cases of the normal source distributions, and experimental results are given for more general distributions, such as the mixtures of uniform distributions. The experiments on real image data sets show that the images sharing the same latent code obtained by diffusion have the similar high level texture features, postures and semantics. Also, the L2 distance seems not suboptimal for visual data. The work bridges the optimal transport theory and the diffusion model.

The contributions are:

1. theoretically verify that for the case of multivariate normal distributions the Monge optimal transport map coincides with the DDPM encoder map.

2. Numerically verification of the similarity between the optimal transport maps and the DDPM encoder map by solving the Fokker-Planck equation on more general synthetic distributions.

3. Additional qualitative empirical evidence supporting the hypothesis on real image datasets are provided.

**Summary Of The Review:**

This work hypothesizes that the limit maps produced by   DDPMs  coincide with the optimal transport maps, or they are very close to each other with high similarity. The work theoretically verifies the hypothesis for the case of multivariate normal distributions the Monge optimal transport map coincides with the DDPM encoder map, gives the numerical verification of the similarity between the optimal transport maps and the DDPM encoder map by solving the Fokker-Planck equation on more general synthetic distributions, and offers additional qualitative empirical evidence supporting the hypothesis on real image datasets are provided.

The idea of bridging optimal transport theory and DDPM is inspiring and important. The mathematical proof is rigorous and elegant, the empirical experimental results are encouraging. The hypothesis can be further formulated to be more quantitively  precise. The mathematical proofs can be directly generalized to other well known distributions. More complicated synthesized distributions can be tested for the hypothesis.

---

> ### Author Response · Authors · 2022-11-18
> **The theoretic proofs can be further improved ...**
>
> We agree that a better explanation at this point will be helpful. We have added it to the updated version of the manuscript. We also have highlighted the connection with the commonly used metric Frechet Inception Distance.

---

> ### Author Response · Authors · 2022-11-18
> **... approximation errors. This need to be carefully analyzed ...**
>
> We used the exact solver for computing the optimal transport available in the POT library - OT Network Simplex Solver. The maximal number of iterations in the solver was set to a sufficiently large number (500 000) so we reached the convergence of the algorithm in all cases.

---

> ### Author Response · Authors · 2022-11-18
> **... the DDPM solution and the optimal transport map are close with small deviation ...**
>
> We agree that a refined formulation of this hypothesis in terms of certain properties of the initial density would be beneficial, however, for now we leave it to future work.

---

### Official Review · Reviewer_6Sgh · 2022-11-03

**Confidence:** 4
**Correctness:** 4
**Technical Novelty And Significance:** 3
**Empirical Novelty And Significance:** 3
**Recommendation:** 8

**Clarity, Quality, Novelty And Reproducibility:**

The paper is written very clearly. The basis of the work, namely recasting diffusion models into continuous SDEs and ODEs and relating flows to transport, is not novel. However, placing everything into this coherent hypothesis and providing clear evidence in support of it is novel, to the best of my knowledge and it is done well. The results appear to be easily reproducible.

**Strength And Weaknesses:**

### Strengths
* Theoretical understanding of denoising diffusion processes is important to bring their impressive qualitative performance with images to other applications. As such, the characterization put forth by this paper can give a good starting point to understanding the nature of the resulting maps.
* The paper is written well, with good overview of the background material, a clear motivation, and directly appreciable results.

### Weaknesses
* The analytical result is very limited, as it pertains to mapping from Gaussians to Gaussians. At some level it is also unsurprising, considering that Gauss-Markov processes do have energy-optimality properties. This doesn’t mean that the result is trivial, but it begs for additional complementation with empirical results to support the hypothesis.
* There is a little bit of ambiguity when moving from the discrete processes to their corresponding continuous versions. For example, the difference equation (4) is a backward process, but the corresponding differential equation (6) appears to be a forward one. It is worth saying something about this to ease reading for those who are not immersed in this area.
* The indirect evidence based on texture similarity in the image data experiments is a very compelling approach. However, it is worth discussing why other reasonable explanations may not work. For example, if the neural network is regularized in all cases to be a Lipschitz map, then the texture similarity could be simply by virtue of that limited distortive ability instead of some global form of transport optimality (though, admittedly, these two concepts could be related.)
* The numerical results paragraph in the image data experiments (p.9) is unclear. To have the analogue of Section 3.4, I would have expected computing the cost of the Monge map for these images, but it doesn’t seem that’s what is done. A clarification here is necessary.


**Summary Of The Paper:**

This paper provides theoretical and experimental support to the following hypothesis: maps induced by denoising diffusion processes are optimal transport maps between the base and target distributions. For the case of multidimensional Gaussians, the paper provides a complete analytical result. For the general case, it shows this directly for synthetic experiments with mixtures of Gaussian and indirectly, through consequences such as texture preservation, for image data.

**Summary Of The Review:**

The paper gives compelling evidence that links denoising diffusion processes to optimal transport maps from base to target distributions. This is done clearly and methodically. This may form the basis of a better understanding of the remarkable performance of this approach to generating natural images and of furthering its use in novel applications. Apart from a couple of clarifications, the paper seems worthwhile to share with the community.

---

> ### Author Response · Authors · 2022-11-18
> **The analytical result is very limited ...**
>
> Thank you for your comments and appreciation of our work! We believe that our analytical result for the Gaussian distribution will inspire other researchers to obtain general proof of the hypothesis. At the same time, we carried out extensive numerical experiments for various non-Gaussian distributions, confirming the conjecture.

---

> ### Author Response · Authors · 2022-11-18
> **... moving from the discrete processes to their corresponding continuous versions ...**
>
> Thank you for this note. We moved the description of the tensor-based Fokker-Planck solver to the appendix (and expand it according to the suggestion of one of the reviewers) and used the vacated space for a more detailed description of the discrete/continuous transformation.

---

> ### Author Response · Authors · 2022-11-18
> **The indirect evidence based on texture similarity ...**
>
> We agree that other explanations are, in principle, possible. For a similar experiment with a conditional GAN model (using the same latent code and decode with two different labels) we did not obtain resembling results, and the produced images were completely different; however, similar arguments about Lipschitz properties would still hold. This suggests that the obtained texture similarities are a 'feature' of diffusion models. Still this is only an indirect evidence and hence we test it more rigorously in numerical results paragraph (p.9).

---

> ### Author Response · Authors · 2022-11-18
> **... paragraph in the image data experiments (p.9) is unclear ...**
>
> For this experiment, we replicated the setup of experiments in Section 3.4 in the following manner. For each dataset we inverted images from the holdout set with the probability flow ODE with the right-hand side represented by a neural network, providing a direct analog of the previous experimental setup. We then compute OT in the same way between the set of images and their latent codes (using l2 distance in pixel space). In all the cases we obtained the identity matrix and transport costs were equal up to machine precision. We did not provide tables with these results in order not to clutter the text. We elaborated on this in the new version of the manuscript.

---

### Decision · Program_Chairs · 2023-01-20

**Decision:**

Accept: poster

**Justification For Why Not Higher Score:**

The theoretical results are a bit limited as it applies only on Gaussian measures. While it  is a more difficult problem, extending those results to other measures (eg mixture of Gaussians) would have make the  theoretical part far stronger.


**Justification For Why Not Lower Score:**

The paper provides useful and novel insights on diffusion model and their codes.

**Metareview: Summary, Strengths And Weaknesses:**


This paper analyzes one bridge between optimal transport and diffusion model.
Its main claim is that diffusion-based models implement transportation plans between target and Gaussian distributions.
While some theoretical support is provided (for some specific situations), the paper supports the
claim throung experimental analysis.

Most reviewers concur on the fact that the paper provides some significant contributions and can be accepted
for publication. However, I urge the authors to improve the final version based on the reviewers ( eg qZHA and R2Hy)
comments.




**Note From Pc:**

if the above contains the word "oral" or "spotlight" please see: "oral" presentation means -> notable-top-5% and "spotlight" means -> notable-top-25%. As stated in our emails, we are disassociating presentation type from AC recommendations